# Emergence of highly pathogenic H5N2 and H7N1 influenza A viruses from low pathogenic precursors by serial passage *in ovo*

**Agnes Tinuke Laleye**[1,2], **Celia Abolnik**[2]*

**1** National Veterinary Research Institute, Vom, Nigeria, **2** Department of Production Animal Studies, Faculty of Veterinary Science, University of Pretoria, Pretoria, South Africa

* celia.abolnik@up.ac.za

**Data Availability Statement:** Raw sequence reads for each passage were deposited in the Genbank BioSample database with BioProject number PRJNA629435 with accession numbers SAMN14775585 - SAMN14775604.

## Abstract

Highly pathogenic (HPAI) strains emerge from their low pathogenic (LPAI) precursors and cause severe disease in poultry with enormous economic losses, and zoonotic potential. Understanding the mechanisms involved in HPAI emergence is thus an important goal for risk assessments. In this study ostrich-origin H5N2 and H7N1 LPAI progenitor viruses were serially passaged seventeen times in 14-day old embryonated chicken eggs and Ion Torrent ultra-deep sequencing was used to monitor the incremental changes in the consensus genome sequences. Both virus strains increased in virulence with successive passages, but the H7N1 virus attained a virulent phenotype sooner. Mutations V63M, E228V and D272G in the HA protein, Q357K in the nucleoprotein (NP) and H155P in the neuraminidase protein correlated with the increased pathogenicity of the H5N2 virus; whereas R584H and L589I substitutions in the polymerase B2 protein, A146T and Q220E in HA plus D231N in the matrix 1 protein correlated with increased pathogenicity of the H7N1 virus in embryos. Enzymatic cleavage of HA protein is the critical virulence determinant, and HA cleavage site motifs containing multibasic amino acids were detected at the sub-consensus level. The motifs PQERRR/GLF and PQRERR/GLF were first detected in passages 11 and 15 respectively of the H5N2 virus, and in the H7N1 virus the motifs PELPKGKK/GLF and PELPKRR/GLF were detected as early as passage 7. Most significantly, a 13 nucleotide insert of unknown origin was identified at passage 6 of the H5N2 virus, and at passage 17 a 42 nucleotide insert derived from the influenza NP gene was identified. This is the first report of non-homologous recombination at the HA cleavage site in an H5 subtype virus. This study provides insights into how HPAI viruses emerge from low pathogenic precursors and demonstrated the pathogenic potential of H5N2 and H7N1 strains that have not yet been implicated in HPAI outbreaks.

## Introduction

Wild aquatic birds are the natural reservoirs of all avian influenza virus (IAV) subtypes that are designated by the combination of hemagglutinin (HA; H1-H16) and neuraminidase (NA;

**Funding:** The study was funded by the South African Exotic Leather Cluster grant "Healthy Flocks-Quality Leather". CA is funded by NRF-DSI SARChI grant #N77705-114612.

**Competing interests:** The authors have declared no competing interests

N1-N9) glycoprotein antigens on the virion [1]. IAVs are further distinguished by their pathogenicity in chickens; most strains are low pathogenic (LPAI) and cause sub-clinical or mild infections, but some LPAI viruses of the H5Nx or H7Nx subtypes (where x denotes any of the nine N subtypes) can mutate to a highly pathogenic form (HPAI) that causes severe disease in poultry, with enormous economic losses and zoonotic potential [2]. The pathogenicity of HPAI viruses is a multigenic trait, but the HA protein plays the major role. HA mediates the attachment of the virus to cellular receptors and viral penetration of the host cell through fusion of the viral envelope with cellular membranes [3, 4]. To facilitate this, the precursor HA protein, $HA_0$, must be proteolytically cleaved into the disulphide linked $HA_1$ and $HA_2$ polypeptides to expose the membrane fusion peptide [5]. However, whether or not proteolytic activation occurs depends on the amino acid motif at the $HA_1$-$HA_2$ cleavage site (HACS) and the presence of an appropriate host endoprotease. Typical LPAI viruses contain the $HA_0$ motifs PQRETR/GLF for H5 or PELPKGK/GLF for H7, i.e. single or non-consecutive basic amino acids (R or K) adjacent to the cleavage site, which are recognised by trypsin-like enzymes that are only expressed in epithelial cells. Thus, the activation and spread of LPAI viruses is confined to the epithelial linings of the respiratory and gastrointestinal tracts. HPAI viruses contain two or more consecutive basic amino acids preceding the cleavage site that are cleavable by subtilisin-like enzymes (e.g. furin) found in a wider range of cell types. Consequently, HPAI viruses replicate in and spread between multiple tissues producing a more severe disease [6, 7].

The conversion of LPAI to HPAI through amino acid modifications at the HACS only occurs in terrestrial poultry, seemingly after a period of circulation in high density populations over the course of a few weeks up to several months [2]. For H5Nx subtypes, the HACS mutation can occur by substitution of nonbasic with basic amino acids, but more frequently by the insertion of additional basic amino acids at the CS, likely through polymerase duplication or slippage events during RNA replication [8–10]. All H7Nx HPAI viruses have insertions of up to ten additional basic amino acids at the HACS, either through RNA polymerase slippage/duplication events or alternatively by non-homologous recombination events with either viral or cellular RNA sequences [11–13]. Homologous recombination in the HACS has not been described in H5Nx viruses. The reasons for why only H5Nx and H7Nx viruses and not the other subtypes are prone to mutations at the HACS, and precise mechanism/s underlying this remain unresolved.

Numerous laboratory studies have emulated the natural mutation of LPAI viruses to HPAI via passages in chickens of different ages and inoculation routes, cell cultures, embryonated chicken eggs and a combination of the aforementioned, with varying success [14]. Embryonated chicken eggs (ECEs) are used extensively in influenza research, routine diagnostic tests, and to propagate viruses for vaccines [15, 16]. Typically, IAVs are inoculated into the allantoic cavities of ECEs aged between 9 and 11 days, when the embryo is approximately halfway through gestation. LPAIVs replicate in the allantoic epithelium where trypsin-like proteases are present and viruses are released into the allantoic fluid, but when inoculated via the chorioallantoic membrane (CAM) route, LPAIV replication is restricted to the endoderm that is composed of allantoic epithelium. In contrast, replication of HPAIVs is unrestricted irrespective of the ECE inoculation route, occurring in multiple cell types including the embryo itself and the blood vessels [17, 18]. Notably, serial passage of LPAIVs in ECEs of 9-to-11 days of age does not lead to the mutation or selection of HPAI viruses in the sub-population, however fourteen-day-old embryos appear to exert a mutational and/or selective pressure enabling HPAIV emergence [19–23]. Perdue and coworkers (1990) first reported that LPAIVs and HPAIVs could be distinguished by their mean death times in older age embryos (12-13-day-old ECEs) but not younger aged embryos (8-9-day-old ECEs), and since then the pathogenic

potential of HPAIVs generated by serial passage in 14-day old ECEs has been assessed by the subsequent intravenous inoculation of chickens and/or the ability to form plaques in cell cultures without trypsin [19–22] or ultra-deep sequencing [14]. The phenomenon of HPAI emergence via passage in 14-day-old ECEs is thought to be facilitated by several factors. The change in anatomy of the growing embryo after day 10 could impede the distribution of LPAIVs because of limited access to trypsin-like proteases [24]; additionally, the increased protease activity in the allantoic fluid of older embryos causes degradation of both LPAIVs and HPAIVs, but HPAIVs could have a selective advantage because they are able to replicate in other anatomical sites such as the embryo itself [19, 25, 26].

In view of the human health threat and economic devastation that HPAI is capable of, understanding how and under which conditions HPAI viruses emerge in poultry from their LPAI progenitors is an important goal for risk assessments. *In ovo* passage in 14-day-old ECEs is a more practical alternative to using live hatched chickens in terms of turnaround time and space required, therefore we conducted seventeen serial passages in 14-day ECEs of H5N2 and H7N1 LPAIV progenitors that were originally isolated from commercial ostriches in order to gain a better understanding of the emergence of HPAIVs. Two LPAI strains, A/ostrich/South Africa/ORD/2012 (H7N1) and A/ostrich/South Africa/325863/2015 (H5N2) were isolated from asymptomatic commercial ostriches (*Struthio camelus*) in the southern Cape region in 2012 and 2015 respectively. In the latter case, prompt and strict control measures prevented spread of the virus, unlike previous H5N2 strains that circulated in flocks and eventually mutated to HPAI [27–30]. Interestingly, no mutation in ostriches of LPAI H7Nx strains to HPAI has ever been reported, despite frequent detection and extended periods of circulation [30]. Ion Torrent ultra-deep sequencing and mean death times were used to monitor the incremental changes in viral genomic molecular markers and the pathotypes, respectively.

## Materials and methods

### Viruses and passage experiments

Experiments in ECEs were conducted with the approval of the Research and Animal Ethics Committees of the University of Pretoria (protocol number V010-17). Second passage stocks of A/ostrich/South Africa/ORD/2012 (H7N1) and A/Ostrich/South Africa/325863/2015 (H5N2) were propagated in 9- to 11- day old specific pathogen free (SPF) ECEs (AviFarms (Pty) Ltd, Pretoria) following the standard international protocol (OIE, 2015), with titration to determine the 50% egg infectious dose ($EID_{50}$) [31].

The allantoic cavities of 3 to 5 14-day old ECEs were inoculated with 0.2 ml of allantoic fluid (AF) containing $10^{6.0}$ $EID_{50}$/0.1 ml of the H5N2 or H7N1 viruses and were incubated for up to 5 days with candling every 24 hours. Eggs containing dead embryos within the first 24 hours of inoculation were discarded as nonspecific mortalities. Thereafter, eggs with dead embryos were removed and kept in a refrigerator at 4°C. All eggs remaining at the end of the incubation period were chilled overnight at 4°C. Daily mortalities were recorded and were used to calculate mean time to death (MDT). The AF was harvested and tested for hemagglutinating (HA) activity according to the standard international standard method [15]. The presence of IAV was confirmed by real-time reverse transcription PCR using the M-gene targeted oligonucleotides and cycling parameters described by Spackman et al. [32] with VetMAX™-Plus One-Step RT-PCR reagents (ThermoFischer Scientific, Johannesburg) on a StepOnePlus thermal cycler (ThermoFischer Scientific, Johannesburg). The IAV-positive AFs from replicate ECEs were pooled and used to inoculate the subsequent passage. Passages 1 to 7 were performed using only the aspirated AFs but, to ensure the broadest possible representation of the viral population within the embryo, from passage eight onwards the whole embryos were

harvested along with the AF, homogenised in antibiotic solution containing 50 mg gentamycin (Virbac, Centurion, South Africa) and 100 mg enrofloxacin (Bayer, Johannesburg) per litre and centrifuged at 1000 x g for 5 minutes, and the supernatants were pooled and inoculated in the subsequent passage. In total, seventeen passages were performed with the H5N2 and H7N1 viruses. All virus passage experiments were performed in the University's Poultry Biosafety Level 3 facility.

## Ion Torrent sequencing

The first seven passages plus passages 11, 15 and 17 were selected for Ion Torrent sequencing, as well as the original inoculum of H5N2 and H7N1 to confirm the consensus sequences previously deposited in Genbank (accession numbers KY765295-KY765302 and KT777901-KT777908, respectively). Total RNA was extracted from AF using TRIzol® reagent (Gibco, Invitrogen) according to the manufacturer's recommended procedure. RNA pellets resuspended in 40 μl diethylpyrocarbonate (DEPC) treated Milli-Q water were quantified with a Nanodrop® Spectrophotometer (ThermoFischer Scientific, Johannesburg) and shipped on ice to the Stellenbosch University Sequencing Facility for Ion Torrent deep sequencing. RNAs were assessed for RNA integrity scores (RIN) and quantified on a BioAnalyzer 2100 using the RNA 6000 Nano Chip and reagents (Agilent Technologies, Waldbronn, Germany). The Ion Total RNA-Seq Kit v2 was used to convert total RNA into a representative cDNA library for strand specific RNA sequencing on the Ion Torrent™ Ion Proton™ system according to manufacturer's protocol. Briefly, 25 μl of total RNA was concentrated to 10 μl at 37°C for 2 hrs. The 10 μl RNA volume was then fragmented with RNAse III for 2 min at 37°C. The fragmented RNA was purified using the magnetic bead clean-up module and eluted in nuclease-free water. The yield and size of the fragmented RNA was not evaluated due to expected low concentrations. Instead, the full volume of RNA was used for subsequent hybridization and ligation to adapters at 30°C for 1 hour. The RNA, with hybridized adapters, was reverse transcribed to generate single stranded cDNA libraries. The cDNA products were amplified to prepare barcoded cDNA libraries using the Ion Xpress™ RNA-Seq Barcode Kit, purified using the magnetic bead clean-up module and assessed for yield and fragment size distribution using the High Sensitivity DNA Kit and chips on the BioAnalyser 2100 (Agilent Technologies) according to recommended procedures. The libraries were considered sufficient for template preparation and enrichment if <50% of fragments were present in the 50 bp to 160 bp range. The barcoded cDNA libraries were diluted to a target concentration of 80 pM and combined in equimolar amounts for sequencing template preparation using the Ion PI™ Hi Q™ Chef Kit. Enriched, template positive ion sphere particles were loaded onto an Ion PI™ (v3) Chip. Massively parallel sequencing was performed on the Ion Proton™ System using solutions and reagents supplied by the manufacturer and according to the recommended procedure. Flow space calibration and basecaller analysis were performed using standard analysis parameters in the Torrent Suite Version 5.4.0 Software (Thermo Fisher Scientific, Waltham, MA, USA). Sequencing reads with bases of Phred scores ≥ 20 were used for analysis.

## Data analysis

**Consensus genome assembly.** Sequence reads were imported into the CLC Genomics Workbench v7.5.2 (CLC Bio, Qiagen) and the default settings were used to assemble the reads against each of the eight reference IAV genome segments for H5N2 (KY765295-KY765302) or H7N1 (KT777901-KT777908) for each passage. Consensus sequences for the eight gene segments at each passage level were imported into BioEdit v7.1.3.0 [33] for multiple sequence alignments of the translated protein sequences. Abbreviations for the encoded genes are as

follows: polymerase B2 (PB2); polymerase B1 (PB1); polymerase B1- Fragment 2 (PB1-F2); polymerase A (PA); polymerase A- X (PA-X); HA: hemagglutinin (HA); nucleoprotein (NP); membrane protein 1 (M1), membrane protein 2 (M2e), non-structural protein 1 (NS1) and nuclear export protein (NEP). Raw sequence reads for each passage were deposited in the Genbank BioSample database under BioProject number PRJNA629435 with accession numbers SAMN14775585—SAMN14775604.

**Hemagglutinin cleavage site variant analysis.**   Two modified sequence tags (MSTs) each for the H5 HACS and H7 HACS were designed and imported into the CLC Genomics Workbench (Table 1). For each passage the sub-set of reads that mapped against Segment 4 (HA gene) was extracted and used to map against the relevant MST pair. The HACS extends from nucleotides 1009 to 1035 in the H5 HA gene and from nucleotides 997 to 1026 in the H7 HA gene, both starting with proline-encoding CCT codon and ending with the phenyalanine-encoding TTT codon (Table 1). Mapped reads extending up to 100 bp upstream or downstream of the HACS were extracted and exported in FASTA file format. In BioEdit the reads were visually inspected and converted to the reverse complement where necessary. The sequences were aligned using MAFFT online v 7 [34] before manual inspection and editing in Bioedit, retaining for analysis only the reads that spanned the entire cleavage site. Amino acid translations were also performed in BioEdit.

## Results

### Increasing pathogenicity to chicken embryos with subsequent passages of H5N2 and H7N1 LPAIVs

Embryos of the sham-inoculated controls (phosphate-buffered saline, pH 7.4) appeared normal whereas the H5N2- and H7N1-virus infected embryos that died within the 90 hour incubation period showed generalised haemorrhages that are characteristic of HPAI, plus stunting and sparse feather suggestive of arrested embryonic development. All eggs inoculated with H5N2 LPAI virus survived the first three passages (Fig 1) whereas all embryos after P4 were dead with MDTs of 60 hours or less. Mean death times and the proportion of surviving embryos infected with H5N2 virus were variable between P4 and P13. The percentage of live embryos and the MDTs had been decreasing up until passage 7, but in passage 8 the phenotype of both viruses changed with an increase in the percentage of live embryos and MDTs of >90 hours. The only change in the protocol was that stocks were frozen at -80˚C during a University recess; this likely caused a slight drop in the viability of the viruses in passage 8, causing the delayed embryo deaths. Eggs inoculated with the H7N1 LPAI virus had a 100% survival rate for the first passage only. From passages 2 to 9 a proportion of the embryos died with each

**Table 1.  Modified sequence tags used to retrieve reads spanning the hemagglutinin cleavage site.**

| LPAI H5 HACS sequence: | **P** | **Q** | **R** | **E** | **T** | **R** | | | **G** | **L** | **F** |
|---|---|---|---|---|---|---|---|---|---|---|---|
| | CCT | CAA | AGA | CAG | ACA | AGA | --- | --- | GGG | CCT | TTT |
| **H5 HA MST 1:** | CCT | CAA | AGA | AAA | AAA | AAA | AAA | --- | GGG | CCT | TTT |
| **H5 HA MST 2:** | CCT | CAA | AGA | AAA | AAA | AAA | AAA | AAA | GGG | CCT | TTT |
| LPAI H7 HACS sequence: | **P** | **E** | **L** | **P** | | **K** | **G** | **R** | **G** | **L** | **F** |
| | CCC | GAA | CTC | CCA | --- | AAG | GGA | AGA | GGC | CTG | TTT |
| **H7 HA MST 1:** | CCC | GAA | CTC | CCA | AAA | AAG | GGA | AGA | GGC | CTG | TTT |
| **H7 HA MST 2:** | CCC | GAA | CTC | CCA | --- | AAA | AAA | AAA | GGC | CTG | TTT |

Amino acid translations are in boldface; changes are underlined; "¬¬-"= no nucleotide.

| Eggs | (a)  H5N2 virus passage No. | | | | | | | | | | | | | | | | |
|---|---|---|---|---|---|---|---|---|---|---|---|---|---|---|---|---|---|
| | 1 | 2 | 3 | 4 | 5 | 6 | 7 | 8 | 9 | 10 | 11 | 12 | 13 | 14 | 15 | 16 | 17 |
| No. inoculated | 3 | 5 | 5 | 5 | 3 | 4 | 4 | 4 | 4 | 3 | 3 | 3 | 3 | 3 | 3 | 4 | 4 |
| No. dead | 0 | 0 | 0 | 2 | 3 | 4 | 4 | 0 | 4 | 3 | 1 | 2 | 3 | 3 | 3 | 4 | 4 |
| No. alive | 3 | 5 | 5 | 3 | 0 | 0 | 0 | 4 | 0 | 0 | 2 | 1 | 0 | 0 | 0 | 0 | 0 |
| Percentage live embryos | 100 | 100 | 100 | 60 | 0 | 0 | 0 | 100 | 0 | 0 | 67 | 33 | 0 | 0 | 0 | 0 | 0 |
| Mean death time in hours | >90 | >90 | >90 | >90 | 60 | 72 | 72 | >90 | 84 | 60 | >90 | >90 | 72 | 48 | 36 | 60 | 48 |

| Eggs | (b)  H7N1 virus passage No. | | | | | | | | | | | | | | | | |
|---|---|---|---|---|---|---|---|---|---|---|---|---|---|---|---|---|---|
| | 1 | 2 | 3 | 4 | 5 | 6 | 7 | 8 | 9 | 10 | 11 | 12 | 13 | 14 | 15 | 16 | 17 |
| No. inoculated | 3 | 5 | 5 | 5 | 3 | 4 | 4 | 4 | 4 | 3 | 3 | 3 | 3 | 3 | 3 | 4 | 4 |
| No. dead | 0 | 2 | 2 | 3 | 3 | 3 | 4 | 1 | 4 | 3 | 3 | 3 | 3 | 3 | 3 | 4 | 4 |
| No. alive | 3 | 3 | 3 | 2 | 0 | 1 | 0 | 3 | 0 | 0 | 0 | 0 | 0 | 0 | 0 | 0 | 0 |
| Percentage live embryos | 100 | 60 | 60 | 40 | 0 | 25 | 0 | 75 | 0 | 0 | 0 | 0 | 0 | 0 | 0 | 0 | 0 |
| Mean death time in hours | >90 | >90 | >90 | >90 | 24 | >90 | 36 | >90 | 72 | 48 | 60 | 60 | 48 | 60 | 36 | 36 | 36 |

**Fig 1.** Mortality patterns and mean death times (MDTs) of (a) H5N2 and (b) H7N1 low pathogenic avian influenza viruses passaged in 14-day old embryonated eggs. A 100% survival rate and MDTs >90 hours are coloured green. Survival rates between 50 and 100% and MDTs between 60 and 90 hours are coloured orange. Survival rates less than 50% and MDTs of 60 hours and less are coloured red.

passage but most had MDTs of 72 hours and above except P5. From P10 onwards all H7N1-infected embryos were dead within 60 hours.

## Emergence of potential virulence markers in the consensus sequences of H5N2 and H7N1 viruses during passages

The total number of sequencing reads generated for the H5N2 virus varied from 3,9 million for P4 to 16,7 million for P11 with average read lengths of between 61 and 138 bp (S1 Table). H5N2 influenza A -specific genome coverage ranged from 8,113 reads mapped to segment 8 in P2 up to 376,580 reads mapped to segment 1 in P11. For H7N1, total read numbers ranged from 3,6 million for P1 to 19,4 million for passage 7 with average read lengths of 91 to 141 bp (S2 Table). H7N1 influenza A -specific genome coverage ranged from 2,196 for segment 2 of P4 up to 526,100 reads mapped to segment 7 in P15.

**Molecular markers in the H5N2 consensus sequences.**   The H5N2 consensus sequences for the PB2, PB1, PB1-F2, PA-X, M2 and NEP proteins did not change over the course of seventeen successive passages in 14-d ECEs (Table 2). A single D216Y substitution emerged in the PA protein after P1 and was retained to P17.

The H5N2 HA protein had the highest proportion of amino acid substitutions. V452I and S511N emerged early in P1 and A200D emerged in P2, but the MDTs were all <90 hours with 100% survival in the first three passages of the virus therefore these three mutations were unlikely to have caused the increased pathogenicity of the virus. Candidates for virulence markers in the HA protein included E228V that appeared in P6 plus V63M and D272G that were present in the consensus sequence from P15 onwards. V63M, E228V and D272G have not previously been experimentally verified as molecular markers involved in H5 AIV pathogenicity, receptor binding, replicative capacity or inter-species transmission [35] nor are present in field HPAIVs to date [36], but E228V is strategically located in the 220-loop of the HA receptor binding site. Alone or in combination, these three substitutions in the HA of the H5N2 virus could be molecular virulence determinants for the increased pathogenicity of the virus from P14 onwards (Fig 1), but experimentally verification is required.

In the NP protein a Q357K mutation emerged in passage 6, and in mice a Q357L mutation (in combination with E627K in PB2) was associated with enhanced virulence [35].

**Table 2. Amino acid changes in the consensus H5N2 virus genomes across passages (P), compared to the original LPAI H5N2 virus.**

| Protein | P1 | P2 | P3 | P4 | P5 | P6 | P7 | P11 | P15 | P17 |
|---|---|---|---|---|---|---|---|---|---|---|
| PB2 | - | - | - | - | - | - | - | - | - | - |
| PB1 | - | - | - | - | - | - | - | - | - | - |
| PB1-F2 | - | - | - | - | - | - | - | - | - | - |
| PA | D216Y | D216Y | D216Y | D216Y | D216Y | D216Y | D216Y | D216Y | D216Y | D216Y |
| PA-X | - | - | - | - | - | - | - | - | - | - |
| HA | - | - | - | - | - | - | - | - | V63M | V63M |
|  | - | A200D | A200D | A200D | A200D | A200D | A200D | A200D | A200D | A200D |
|  | - | - | - | - | - | E228V | E228V | E228V | E228V | E228V |
|  | - | - | - | - | - | - | - | - | D272G | D272G |
|  | V452I | V452I | V452I | V452I | V452I | V452I | V452I | V452I | V452I | V452I |
|  | S511N | S511N | S511N | S511N | S511N | S511N | S511N | S511N | S511N | S511N |
| NP | - | - | - | - | - | Q357K | Q357K | Q357K | Q357K | Q357K |
| NA | - | - | - | - | - | - | - | H155P | H155P | H155P |
|  | I464N | I464N | I464N | I464N | I464N | I464N | I464N | - | - | - |
| M1 | - | T168I | T168I | T168I | T168I | T168I | T168I | T168I | T168I | T168I |
| M2 | - | - | - | - | - | - | - | - | - | - |
| NS1 | - | - | - | - | - | - | - | E60G | E60G | E60G |
| NEP | - | - | - | - | - | - | - | - | - | - |

- No change.

In the NA protein, I464N emerged in P1, but disappeared from the consensus sequence after P7. An H155P substitution detected in P11 was retained to P17, but neither of these molecular markers is known to be associated with increased pathogenicity or resistance to antiviral compounds. Similarly, the T168I substitution in the M1 protein from P2 onwards was not previously described as virulence determinants [35, 36]. The E60G substitution in NS1 was however also present in Mexican HPAI H5N2 strains from 1994 to 1995 that emerged from low pathogenic precursors [37].

**Molecular markers in the H7N1 consensus sequences.** Two amino acid substitutions in the H7N1 consensus PB2 proteins, namely P349L and H748Q, plus P58L in the NEP emerged in P1 (Table 3) where MDTs were >90 hours (Fig 1) and were therefore unlikely to influence the pathogenicity of the H7N1 strain. All eggs had died consistently from P10 onwards and in P11, when PB1 R584H and L589I had emerged in the consensus genome. These two candidates for virulence markers in the PB1 protein have not been described before.

The H7N1 HA protein had several substitutions appearing from P6 onwards. A146T emerged between P8 and P15 where it was present in the consensus sequence of latter passage. The A146T mutation was also observed in an H7N3 HPAI that emerged from low pathogenic precursor in Italy [38], although the significance in host adaptation and pathogenicity shift is yet to be elucidated. A146S was one of four HA mutations in an H7N7 virus found to be associated with increased pathogenicity in mice and transmission in guinea pigs [39]. The Q220E and N455Y substitutions also appeared post P7 but were not stably maintained in the majority viral population. The amino acid sequence in the M1 protein is characterized by a pair of aspartate residues at positions 231 and 232. In this study D231N in the M1 protein detected in P15 was retained until the end of the experiment at passage 17 but interestingly, only the adjacent D232N is associated with field HPAIVs [35].

**Table 3. Amino acid changes in the consensus H7N1 genome across passages (P), compared to the original LPAI H7N1 virus.**

| Protein | P1 | P2 | P3 | P4 | P5 | P6 | P7 | P11 | P15 | P17 |
|---|---|---|---|---|---|---|---|---|---|---|
| PB2 | P349L | P349L | P349L | P349L | P349L | P349L | P349L | P349L | P349L | P349L |
| | H748Q | H748Q | H748Q | H748Q | H748Q | H748Q | H748Q | H748Q | H748Q | H748Q |
| PB1 | - | - | - | - | - | - | - | R584H | R584H | R584H |
| | - | - | - | - | - | - | - | L589I | L589I | L589I |
| PB1-F2 | - | - | - | - | - | - | - | - | - | - |
| PA | - | - | - | - | - | - | - | - | - | - |
| PA-X | - | - | - | - | - | - | - | - | - | - |
| HA | - | - | - | - | - | - | - | - | A146T | A146T |
| | - | - | - | - | - | - | - | Q220E | Q220E | - |
| | - | - | - | - | - | N434K | - | - | - | - |
| | - | - | - | - | - | - | N455Y | N455Y | - | - |
| NP | - | - | - | - | - | - | - | - | - | - |
| NA | - | - | - | - | - | S369G | - | - | - | - |
| M1 | - | - | - | - | - | - | - | - | D231N | D231N |
| M2e | - | - | - | - | - | - | - | - | - | - |
| NS1 | - | - | - | - | - | - | - | - | - | - |
| NEP | P58L | P58L | P58L | P58L | P58L | P58L | P58L | P58L | P58L | P58L |

- No change.

## Variation within the HACS and emergence of multibasic motifs of H5N2 and H7N1 viruses

Modified sequence tags (MSTs) (Table 1) were used to extract all reads completely spanning the HACS of the H5N2 and H7N1 viruses. Altering the MSTs to include longer insertions of A or G nucleotides, and modifying nucleotides at other positions were tested but these did not result in the retrieval of longer inserts or other populations. Reads were sorted according to length and translated to amino acids.

**Variation at the HACS of the H5N2 virus during passage.** The total number of reads recovered for the HACS of the H5N2 virus varied from 40 for P2 to 425 for P6 (S3 Table). Variation at the HACS was generated through random deletions of between one and three nucleotides, insertions of up to six nucleotides and substitutions, resulting in amino acid substitutions and/ or frame shifts in the encoded peptides. There was no correlation between the number of HACS reads recovered and the number of variants detected. The original LPAI nucleotide sequence dominated the total read percentage at each passage level, with the lowest frequency of 47.71% at P15. Overall, there appeared to be a general increase in the proportion of variant HACS sequences after P6, but there was no increase in the frequency of longer insertions. Notably, non-homologous recombination in the HACS was evident in two instances; in P6 an insertion of 13 nts encoding the peptide TGTGV (not in frame) originated from an unknown source and in P17 a 42-nt insert in-frame with and immediately adjacent to the HACS was found. The insert (underlined) encoded the motif PQRETRRESRNPGNAEIEDLIF and bore 100% identity similarity with nucleotides 729 to 776 of the NP gene of influenza A virus [40].

**Detection of in-frame monobasic and multibasic HACS motifs in the H5N2 virus across passages.** H5 HACS sequences that encoded complete HACS motifs flanked by "PQ" and "GLF" residues (S3 Table) are collated in Table 4. The motif PQRGTR/GLF (variation is underlined), detected in P1, P11 and P15 at low frequency is commonly detected in wild bird

**Table 4. Hemagglutinin cleavage sites (HA$_0$) detected in the H5N2 low pathogenic avian influenza virus passaged in 14-day old ECEs.**

| Passage no. (total reads) | HA$_0$ cDNA nucleotide sequence | Number of reads (percentage) | Translated amino acid | Predicted cleavability *in vivo* |
|---|---|---|---|---|
| **1** (225) | CCTCAAAGAGAGACAAGAGGGCTATTT | 184 (82.14) | **PQRETR/GLF** | conventional monobasic LPAI |
| | CCTCAAAGAGGGCAAGAGGGCTATTT | 2 (0.89) | **PQRGTR/GLF**[1] | monobasic, LPAI |
| | CCTCAAAGACGAGACA_GAGGGCTATTT | 1 (0.44) | **PQRRDR/GLF** | monobasic, LPAI |
| **2** (40) | CCTCAAAGAGAGACAAGAGGGCTATTT | 33 (82.50) | **PQRETR/GLF** | conventional monobasic LPAI |
| **3** (57) | CCTCAAAGAGAGACAAGAGGGCTATTT | 46 (80.70) | **PQRETR/GLF** | conventional monobasic LPAI |
| | CCTCAAAAAGAGACAAGGGGGCTATTT | 1 (1.75) | **PQKETR/GLF**[2] | monobasic, LPAI |
| **4** (259) | CCTCAAAGAGAGACAAGAGGGCTATTT | 240 (92.66) | **PQRETR/GLF** | conventional monobasic LPAI |
| | CCTCAAAGAGAGGCAAGAGGGCTATTT | 2 (0.77) | **PQREAR/GLF**[3] | monobasic, LPAI |
| | CCTCAAAGAGAGACAAAAGGGCTATTT | 1 (0.39) | **PQRETK/GLF**[4] | monobasic, LPAI |
| | CCTCAAACGAGAGCACCAAGAGGGCTATTT | 1 (0.39) | **PQTRAPR/GLF** | monobasic, LPAI |
| **5** (290) | CCTCAAAGAGAGACAAGAGGGCTATTT | 267 (92.07) | **PQRETR/GLF** | conventional monobasic LPAI |
| **6** (425) | CCTCAAAGAGAGACAAGAGGGCTATTT | 404 (95.06) | **PQRETR/GLF** | conventional monobasic LPAI |
| | CCTCAA_GAGAGACAACGAGGGCTATTT | 2 (0.47) | **PQERQR/GLF** | monobasic, LPAI |
| | CCTCAAAAAGAGACAAGAGGGCTATTT | 1 (0.24) | **PQKETR/GLF**[2] | monobasic, LPAI |
| **7** (114) | CCTCAAAGAGAGACAAGAGGGCTATTT | 82 (71.93) | **PQRETR/GLF** | conventional monobasic LPAI |
| | CCTCAA_GAGAGGACAAGAGGGCTATTT | 1 (0.88) | **PQERTR/GLF** | monobasic, LPAI |
| **11** (208) | CCTCAAAGAGAGACAAGGGGGCTATTT | 134 (64.42) | **PQRETR/GLF** | conventional monobasic LPAI |
| | CCTCAAGAGAGACGAAGAGGGCTATTT | 1 (0.48) | **PQERRR/GLF** | **multibasic, HPAI** |
| | CCTCAAAGAGAAGACA_GAGGGCTATTT | 1 (0.48) | **PQREDR/GLF** | monobasic, LPAI |
| | CCTCAAAGAGGGCAAGAGGGCTATTT | 1 (0.48) | **PQRGTR/GLF**[1] | monobasic, LPAI |
| | CCTCAAAGAAGAGGACAAGAGGGCTTATTT | 3 (1.44) | **PQRRGQE/GLF** | monobasic, LPAI |
| **15** (301) | CCTCAAAGAGAGACAAGAGGGCTATTT | 146 (47.71) | **PQRETR/GLF** | conventional monobasic LPAI |
| | CCTCAAAAAGAGACAAGAGGGCTATTT | 2 (0.66) | **PQKETR/GLF**[2] | monobasic, LPAI |
| | CCTCAAAAGAAGAGACAAGACGGGCTATTT | 1 (0.33) | **PQKKRQD/GLF** | monobasic, LPAI |
| | CCTCAAAGAGAGGCAAGAGGGCTATTT | 3 (0.98) | **PQREAR/GLF**[3] | monobasic, LPAI |
| | CCTCAAAGAGAGAGAAGAGGGCTATTT | 3 (0.98) | **PQRERR/GLF** | **multibasic, HPAI** |
| | CCTCAAAGAGGGCAAGAGGGCTATTT | 3 (0.98) | **PQRGTR/GLF**[1] | monobasic, LPAI |
| | CCTCAAAGAAAGACAAGAGGGCTATTT | 1 (0.33) | **PQRKTR/GLF**[5] | monobasic, LPAI |
| | CCTCAAAGAAGAGACGAAGAAGGGCTATTT | 1 (0.33) | **PQRRDEE/GLF** | monobasic, LPAI |
| | CCTCAAAGAAGAGGACAAGAGGGCTTATTT | 1 (0.33) | **PQRRGQE/GLF** | monobasic, LPAI |
| | CCTCAAAGACGTAGACAAGAGGGCTTATTT | 1 (0.33) | **PQRRRQE/GLF** | monobasic, LPAI |
| **17** (130) | CCTCAAAGAGAGACAAGAGGGCTATTT | 81 (62.31) | **PQRETR/GLF** | conventional monobasic LPAI |
| | CCTCAA_GAGAGAGCAAGAGGGCTATTT | 1 (0.77) | **PQERAR/GLF** | monobasic, LPAI |
| | CCTCAA_GAACGAGACAAGATGGGCCTATTT | 1 (0.77) | **PQERDKM/GLF** | monobasic, LPAI |
| | CCTCAA_GAGAGACAACGAGGGCTATTT | 1 (0.77) | **PQERQR/GLF** | monobasic, LPAI |
| | CCTCAA_GAGAGACGAAGAGGGCTATTT | 1 (0.77) | **PQERRR/GLF** | **multibasic, HPAI** |
| | CCTCAAAGAGAGATAAGAGGGCTATTT | 1 (0.77) | **PQREIR/GLF**[6] | monobasic, LPAI |
| | CCTCAAAGAAGAGGACAAGAGGGCTTATTT | 2 (1.54) | **PQRRGQE/GLF** | monobasic, LPAI |
| | CCTCAAAGAGTACGGACGAACGAGGGCCTATTT | 1 (0.77) | **PQRVRTNE/GLF** | monobasic, LPAI |

[1,2,3] Motifs common in wild bird H5Nx LPAIVs.

[4] Motif detected in A/gull/Pennsylvania/4175/83(H5N1) (NCBI Sequence ID: AAD13575) and South Korean wild duck H5N2 LPAIVs.

[5] Motif common in wild bird and poultry H5Nx LPAIVs.

[6] Motif detected in A/Muscovy duck/New York/09-005059-002/2009 (H5N2) (NCBI Sequence ID: AWX60259) and A/duck/New York/09-005059-001/2009 (H5N2) (NCBI Sequence ID: AWX60391).

LPAIVs [40]. The motifs PQKETR/GLF (P3, P6 and P15), PQREAR/GLF (P4 and P15), and PQRKTR/GLF (P15) were detected at low frequencies and are commonly detected in wild bird and poultry LPAIVs [40]. PQRETK/GLF detected in P4 has been previously described in A/gull/Pennsylvania/4175/83 (H5N1) as well as South Korean wild duck H5N2 LPAIVs. PQREIR/GLF in P17 was also previously reported in duck-origin H5N2 LPAIVs from North America, namely A/Muscovy duck/New York/09-005059-002/2009 and A/duck/New York/09-005059-001/2009 [40].

An HACS that contained a dibasic amino acid pair adjacent to the peptide cleavage site was first detected in P11, with the sequence of PQERRR/GLF. The additional arginine residue resulted not from a duplication event, but rather from the deletion of A and the insertion of a G nucleotide, but this sequence has not yet been reported in nature. The same motif was detected in P17 but was not in P15. A second multi-basic motif, viz. PQRERR/GLF, was detected in P15 where the T to R substitution was caused by an A to G nucleotide mutation.

**Variation at the HACS of the H7N1 virus during passage.** Substantially more reads spanning the complete HACS were recovered for the H7N1 virus (S4 Table), even though the depth of coverage obtained for the segment 4 for H7N1 compared to the H5N2 virus was variable (S1 and S2 Tables). Total reads spanning the entire HACS in H7N1 ranged from 60 recovered for P4 to 13,681 recovered for P15. Similar to the H5N2 virus, there was no obvious correlation between the number of HACS reads and the amount of variation detected, but unlike the H5N2 virus there was no apparent increase in the proportion of variants compared to the original LPAI sequence in the later H7N1 virus passages; the original LPAI sequence remained in the vast majority at above 93.96% across passages.

**Detection of in-frame monobasic and multibasic HACS motifs in the H7N1 virus across passages.** H7 HACS sequences that encoded complete HACS motifs flanked by "PE" and "GLF" residues (S4 Table) are collated in Table 5. The motif PELPKGK/GLF was detected in P2, P3, P7, P15 and P17 and was previously described in wild bird LPAIVs in the Netherlands and Korea [40]. PEPPKGR/GLF (P2, P7, P11 and P15) is a common motif in wild bird H7Nx LPAIVs [40]. PEFPKGR/GLF (P3, P5, P7 and P11) was detected in an H7N3 LPAI isolate, A/ruddy shelduck/Mongolia/598C2/2009. PEVPKGR/GLF (P3 only), is a common motif in wild duck and poultry H7N1 and H7N9 LPAIVs (NCBI, 2020). PESPKGR/GLF (P7 and P15) was previously identified in H7N7 strain A/goose/Guangdong/7472/2012 [40], whereas PETPKGR/GLF (P7, P11 and P15) is a common motif in wild bird and poultry H7Nx LPAIVs (NCBI, 2000).

Multi-basic H7N1 HACS motifs were first detected in P7. PELPKGKK/GLF was only detected in P7, and the mutation was the result of insertions of A and G nucleotides at non-consecutive positions in the HACS. This motif has not been reported in nature, but the second multi-basic motif in P7, PELPKRR/GLF, was previously identified in an H7N6 isolate, A/quail/Aichi/4/2009 [41]. Here, the G to R substitution was caused by a G to A mutation in the nucleotide sequence, and this motif was also detected in P15. Even though H7 multi-basic cleavage site sequences were not detected in P11 or P17 (possibly due to insufficient sequencing depth in the specific region) it is likely that they were present according to the consistently low MDTs (Fig 1).

## Discussion

Deep sequencing technologies have revolutionised studies on viral evolution, and here the emergence of highly pathogenic H5N2 and H7N1 avian influenza viruses from low pathogenic precursors was followed over the course of seventeen serial passages in embryonated chicken eggs. The 14-day egg model is a useful alternative to infecting live birds for studying the

**Table 5. Hemagglutinin cleavage sites (HA$_0$) detected in the H7N1 low pathogenic avian influenza virus passaged in 14-day old ECEs.**

| Passage no. (total reads) | HA$_0$ cDNA nucleotide sequence | Number of reads (percentage) | Translated amino acid | Predicted cleavability *in vivo* |
|---|---|---|---|---|
| **1** (1778) | CCCGAACTCCCAAAGGGAAGAGGCCTGTTT | 1690 (95.05) | **PELPKGR/GLF** | conventional monobasic LPAI |
| | CCCGAACTCTCAAAGGGAAGAGGCCTGTTT | 8 (0.45) | **PELSKGR/GLF** | monobasic, LPAI |
| | CCCGAACTCCCAAGGGGAAGAGGCCTGTTT | 7 (0.39) | **PELPRGR/GLF** | monobasic, LPAI |
| | CCCGAACGCCCAAAGGGAAGAGGCCTGTTT | 1 (0.06) | **PERPKGR/GLF** | monobasic, LPAI |
| **2** (2869) | CCCGAACTCCCAAAGGGAAGAGGCCTGTTT | 2730 (95.15) | **PELPKGR/GLF** | conventional monobasic LPAI |
| | CCCGAACTCCCAAAGGAAAGAGGCCTGTTT | 1 (0.04) | **PELPKER/GLF** | monobasic, LPAI |
| | CCCGAACTCCCAAAGGGAAAAGGCCTGTTT | 4 (0.14) | **PELPKGK/GLF**[1] | monobasic, LPAI |
| | CCCGAACCCCCAAAGGGAAGAGGCCTGTTT | 3 (0.11) | **PEPPKGR/GLF**[2] | monobasic, LPAI |
| | CCCGAACTCCGAAAGGGAAGAGGCCTGTTT | 1 (0.04) | **PELRKGR/GLF** | monobasic, LPAI |
| | CCCGAACTCCCAAGGGGAAGAGGCCTGTTT | 5 (0.17) | **PELPRGR/GLF** | monobasic, LPAI |
| **3** (4016) | CCCGAACTCCCAAAGGGAAGAGGCCTGTTT | 3910 (97.36) | **PELPKGR/GLF** | conventional monobasic LPAI |
| | CCCGAACATCCAAAGGGAAGAGGCCTGTTT | 1 (0.03) | **PEHPKGR/GLF** | monobasic, LPAI |
| | CCCGAACTCCCAAAGGGAAAAGGCCTGTTT | 1 (0.03) | **PELPKGK/GLF**[1] | monobasic, LPAI |
| | CCCGAATTCCCAAAGGGAAGAGGCCTGTTT | 1 (0.03) | **PEFPKGR/GLF**[3] | monobasic, LPAI |
| | CCCGAACTCCCAAGGGGAAGAGGCCTGTTT | 7 (0.17) | **PELPRGR/GLF** | monobasic, LPAI |
| | CCCGAACTCCCAACGGGAAGAGGCCTGTTT | 2 (0.05) | **PELPTGR/GLF** | monobasic, LPAI |
| | CCCGAAGTCCCAAAGGGAAGAGGCCTGTTT | 2 (0.05) | **PEVPKGR/GLF**[4] | monobasic, LPAI |
| | CCCGAACTCCAAAAGGGAAGAGGCCTGTTT | 2 (0.05) | **PELQKGR/GLF** | monobasic, LPAI |
| | CCCGAACTCCAACGGAAGGAGGGCCTGTTT | 1 (0.03) | **PELQRKE/GLF** | monobasic, LPAI |
| | CCCGAACTCTCAAAGGGAAGAGGCCTGTTT | 1 (0.03) | **PELSKGR/GLF** | monobasic, LPAI |
| | CCCGAACCCCCAAAGGGAAGAGGCCTGTTT | 1 (0.03) | **PEPPKGR/GLF**[2] | monobasic, LPAI |
| **5** (74) | CCCGAACTCCCAAAGGGAAGAGGCCTGTTT | 72 (97.30) | **PELPKGR/GLF** | conventional monobasic LPAI |
| | CCCGAATTCCCAAAGGGAAGAGGCCTGTTT | 1 (1.35) | **PEFPKGR/GLF**[3] | monobasic, LPAI |
| **6** (1030) | CCCGAACTCCCAAAGGGAAGAGGCCTGTTT | 985 (95.63) | **PELPKGR/GLF** | conventional monobasic LPAI |
| | CCCGAACTCTCAAAGGGAAGAGGCCTGTTT | 1 (0.10) | **PELSKGR/GLF** | monobasic, LPAI |
| | CCCGAACTCCCAAGGGGAAGAGGCCTGTTT | 3 (0.29) | **PELPRGR/GLF** | monobasic, LPAI |
| **7** (2331) | CCCGAACTCCCAAAGGGAAGAGGCCTGTTT | 2232 (95.75) | **PELPKGR/GLF** | conventional monobasic LPAI |
| | CCCGAATTCCCAAAGGGAAGAGGCCTGTTT | 2 (0.09) | **PEFPKGR/GLF**[3] | monobasic, LPAI |
| | CCCGAACATCCCAAGGGAAGAGGCCTGTTT | 1 (0.04) | **PEHPKGR/GLF** | monobasic, LPAI |
| | CCCGAACTCCCAAAGGAAAGAGGCCTGTTT | 1 (0.04) | **PELPKER/GLF** | monobasic, LPAI |
| | CCCGAACTCCCAAAGGGAAAAGGCCTGTTT | 2 (0.09) | **PELPKGK/GLF**[1] | monobasic, LPAI |
| | CCCGAACTCCCAAAGAGAAGAGGCCTGTTT | 3 (0.13) | **PELPKRR/GLF**[5] | **multibasic, HPAI** |
| | CCCGAACTCCCAA_GGGAACGAGGCCTGTTT | 2 (0.09) | **PELPRER/GLF** | monobasic, LPAI |
| | CCCGAACTCCCAAGGGGAAGAGGCCTGTTT | 1 (0.04) | **PELPRGR/GLF** | monobasic, LPAI |
| | CCCGAACTCCAACGGAAGGAAGGCCTGTTT | 1 (0.04) | **PELQRKE/GLF** | monobasic, LPAI |
| | CCCGAACCCCCAAAGGGAAGAGGCCTGTTT | 3 (0.13) | **PEPPKGR/GLF**[2] | monobasic, LPAI |
| | CCCGAATCCCCAAAGGGAAGAGGCCTGTTT | 1 (0.04) | **PESPKGR/GLF**[6] | monobasic, LPAI |
| | CCCGAAACTCC_AAAGGGAAGAGGCCTGTTT | 1 (0.04) | **PETPKGR/GLF**[7] | monobasic, LPAI |
| | CCCGAACTCCCAAAGGGGAAGAA_GGGCCTGTTT | 1 (0.04) | **PELPKGKK/GLF** | **multibasic, HPAI** |
| | CCCGAACTCCCCAAAAGGGGAAGAGGCCTGTTT | 1 (0.04) | **PELPKRGR/GLF** | monobasic, LPAI |
| | CCCGAAACTCCCCAAAAGGGAAGAGGCCTGTTT | 1 (0.04) | **PETPQKGR/GLF** | monobasic, LPAI |

(*Continued*)

**Table 5.** (Continued)

| Passage no. (total reads) | HA$_0$ cDNA nucleotide sequence | Number of reads (percentage) | Translated amino acid | Predicted cleavability *in vivo* |
|---|---|---|---|---|
| **11 (3625)** | CCCGAACTCCCAAAGGGAAGAGGCCTGTTT | 3406 (93.95) | **PELPKGR/GLF** | conventional monobasic LPAI |
| | CCCGAA**TT**CCCAAAGGGAAGAGGCCTGTTT | 1 (0.03) | **PEFPKGR/GLF**[3] | monobasic, LPAI |
| | CCCGAA**A**TCCCAAAGGGAAGAGGCCTGTTT | 2 (0.06) | **PEIPKGR/GLF**[8] | monobasic, LPAI |
| | CCCGAACTCCCAA**G**GGGAAGAGGCCTGTTT | 3 (0.08) | **PELPRGR/GLF** | monobasic, LPAI |
| | CCCGAACTCCAACGGAAGGAAGGCCTGTTT | 1 (0.03) | **PELQRKE/GLF** | monobasic, LPAI |
| | CCCGAAC**C**CCCAAAGGGAAGAGGCCTGTTT | 4 (0.11) | **PEPPKGR/GLF**[2] | monobasic, LPAI |
| | CCCGAA**A**ACTCCCAAGGGAAGAGGCCTGTTT | 1 (0.03) | **PETPKGR/GLF**[7] | monobasic, LPAI |
| **15 (13681)** | CCCGAACTCCCGAAGGGAAGAGGCCTGTTT | 12985 (94.72) | **PELPKGR/GLF** | conventional monobasic LPAI |
| | CCCGAACTCCC**A**AAGGGAAA**A**AGGCCTGTTT | 5 (0.04) | **PELPKGK/GLF**[1] | monobasic, LPAI |
| | CCCGAACA**C**CCAAAGGGAAGAGGCCTGTTT | 2 (0.02) | **PEHPKGR/GLF** | monobasic, LPAI |
| | CCCGAAA**T**CCCAAAGGGAAGAGGCCTGTTT | 1 (0.01) | **PEIPKGR/GLF**[8] | monobasic, LPAI |
| | CCCGAACTCC**T**AAAGGGAAGAGGCCTGTTT | 3 (0.02) | **PELLKGR/GLF** | monobasic, LPAI |
| | CCCGAACTCCCAAAGG**A**AAGAGGCCTGTTT | 6 (0.04) | **PELPKER/GLF** | monobasic, LPAI |
| | CCCGAACTCC**A**AAAGGGAAGAGGCCTGTTT | 1 (0.01) | **PELQKGR/GLF** | monobasic, LPAI |
| | CCCGAACTCCAA**C**GGAAGGAAGGCCTGTTT | 1 (0.01) | **PELQRKE/GLF** | monobasic, LPAI |
| | CCCGAACTCCCAAAGA**G**AAGAGGCCTGTTT | 1 (0.01) | **PELPKRR/GLF**[5] | **multibasic, HPAI** |
| | CCCGAACTCCCAAAGG**T**AAGAGGCCTGTTT | 7 (0.05) | **PELPKVR/GLF** | monobasic, LPAI |
| | CCCGAACTCCCAA**G**GGGAAGAGGCCTGTTT | 13 (0.10) | **PELPRGR/GLF** | monobasic, LPAI |
| | CCCGAACTCC**G**AAAGGGAAGAGGCCTGTTT | 1 (0.01) | **PELRKGR/GLF** | monobasic, LPAI |
| | CCCGAAC**G**CCCAAAGGGAAGAGGCCTGTTT | 1 (0.01) | **PERPKGR/GLF** | monobasic, LPAI |
| | CCCGAA**A**ACTCCC_AAGGGAAGAGGCCTGTTT | 2 (0.02) | **PETPKGR/GLF**[7] | monobasic, LPAI |
| | CCCGAAC**C**CCCAAAGGGAAGAGGCCTGTTT | 16 (0.12) | **PEPPKGR/GLF**[2] | monobasic, LPAI |
| | CCCGAACTCCCGAAAGGGAAGA**C**AGGCCTGTTT | 1 (0.01) | **PELPKGKT/GLF** | monobasic, LPAI |
| | CCCGAACA**T**CCCAAA**C**GGGAAA**A**GAGGCCTGTTT | 1 (0.01) | **PEHPKRER/GLF** | monobasic, LPAI |
| | CCCGAAA**A**CTCCCCAAAAGGGAAGAGGCCTGTTT | 1 (0.01) | **PETPQKGR/GLF** | monobasic, LPAI |
| | CCCGAA**A**ACTCCCAAAGGGAA**G**GAGGG**CCTGTTT | 1 (0.01) | **PETPKGKE/GLF** | monobasic, LPAI |
| | CCCGAA**A**ACTCCCC**AAAAGGG**GAAAGAGGG**CCTGTTT | 1 (0.01) | **PETPQKGKE/GLF** | monobasic, LPAI |
| **17 (7280)** | CCCGAACTCCCAAAGGGAAGAGGCCTGTTT | 7079 (97.24) | **PELPKGR/GLF** | conventional monobasic LPAI |
| | CCCGAACTCCCAAAGGGAAA**A**GGCCTGTTT | 1 (0.01) | **PELPKGK/GLF**[1] | monobasic, LPAI |
| | CCCGAACTC**T**CAAAGGGAAGAGGCCTGTTT | 2 (0.03) | **PELSKGR/GLF** | monobasic, LPAI |
| | CCCGAACTCCCAA**G**GGGAAGAGGCCTGTTT | 8 (0.11) | **PELPRGR/GLF** | monobasic, LPAI |
| | CCCGAACTCCCAAAGA**G**AGGAGGCCTGTTT | 1 (0.01) | **PELPKRG/GLF** | monobasic, LPAI |
| | CCCGAAA**A**CTCCCAAA**C**GGGAAGAGGGCCTGTTT | 1 (0.01) | **PETPKREE/GLF** | monobasic, LPAI |

[1]Motif detected in wild bird H7N1 LPAIVs in the Netherlands and Korea in 2011, e.g. A/mallard duck/Netherlands/43/2011(H7N1) NCBI Sequence ID: APC30435.

[2, 7, 8] Motifs common in wild bird H7Nx LPAIVs.

[3]Motif detected in A/ruddy shelduck/Mongolia/598C2/2009 (H7N3), NCBI Sequence ID: AGL07608.

[4]Motif detected in wild duck and poultry H7N1 and H7N9 LPAIVs.

[5]Motif detected in A/quail/Aichi/4/2009(H7N6) NCBI Sequence ID: BAJ08819.

[6]Motif detected in A/goose/Guangdong/7472/2012(H7N7) Sequence ID: AGQ80959.

LPAI-HPAI conversion/ selection process [14, 19, 21] as well as viral pathogenicity by the embryo mean death times [8].

The H5N2 and H7N1 precursor strains were isolated from commercial ostriches but were not associated with any HPAI outbreaks, and deep sequencing of the stocks used for the

experiments confirmed that only LPAIVs were present in the sub-populations. Whereas initially both viruses were avirulent with 100% of the embryos surviving and MDTs of >90 hours, both strains gradually increased in virulence for 14-day old chick embryos as the passages progressed. By P14 100% of the embryos were dead, and by the seventeenth passage the MDTs were markedly shorter at 48 h and 36 h for H5N2 and H7N1 respectively. The progression of pathogenicity *in ovo* was markedly different for the two viral strains; the H5N2 virus remained avirulent longer than the H7N1 strain since three passages of H5N2 compared to just one of H7N1 virus were required before the embryos started to die. The H7N1 virus also attained high virulence sooner, with 100% mortalities and MDTs of < 60 hours being reached at P10, four passages earlier than the H5N2 virus.

The HACS is the key virulence determinant but it is not sufficient for expression of full virulence [42, 43] therefore; we monitored the emergence of molecular markers in the various proteins encoded by the consensus sequences. In the H5N2 virus the combination of emergent V63M, E228V and D272G substitutions in the HA, Q357K in NP and H155P in NA correlated with the increased pathogenicity of H5N2 according to the MDT profile. Most interestingly the E60G substitution in NS1 was also present in H5N2 HPAIVs that emerged from an LPAI precursor in a Mexican poultry outbreak [37]. The viral NS1 protein is a well-known virulence factor that suppresses the host's innate immunity by preventing host cell mRNA processing, blocking the nuclear export of polyadenylated cellular transcripts and inhibiting type I interferon responses, as well as an inhibitor of adaptive immunity through its various effects on dendritic cells [44].

The combination of R584H and L589I substitutions in PB2, A146T (and possibly Q220E) in HA with D231N in M1 correlated with the increasing MDTs for the H7N1 virus. Overall, the results for these H5N2 and H7N1 strains are consistent with other studies that compared the switch from LPAI to HPAI, whereby between 7 and 68 amino acids are substituted with changes occurring in the HA gene, but also often in the polymerase genes [37]. Apart from E60G substitution in the NS1 of the H5N2 virus, none of the other mutations in H5N2 or H7N1 detected here were present in the H5Nx and H7Nx strains used in other LPAI-HPAI conversion studies, nor are they known to be associated with increased pathogenicity [14, 35, 37], reflecting the complexity of viral pathogenesis outside of the HACS.

Deep sequencing enables the study of minority variants at the HACS which otherwise may not be identifiable at the consensus level. Ion Torrent sequencing is known to have a high error rate in base calls in long homopolymer regions [45], but extended homopolymeric regions such as those caused by RNA polymerase slippage were absent from our data. In the H5N2 virus an HACS containing a dibasic amino acid pair adjacent to the peptide cleavage site, PQERRR/GLF, was first detected in P11 and a second motif, PQRERR/GLF, was detected in P15. Even though the intervening passages (P8 to P10) were retrospectively sequenced to pinpoint the emergence of the H5N2 multi-basic HACS in the population, the depth of coverage obtained was too poor for analysis, probably due to degradation of the RNA during prolonged storage. Neither of the aforementioned H5 multi-basic motifs has been reported in nature yet [40, 43]. Two multi-basic HACS motifs, PELPKGKK/GLF and PELPKRRGLF, were detected for the first time in P7 of the H7N1 virus but only the latter has been found in nature thus far, in an H7N9 virus isolated from an outbreak in quails in Japan [40, 41].Thus, Fig 2 summarizes that the increasing virulence of the H5N2 and H7N1 viruses was an accretion of the HA multibasic cleavage sites, other amino acid substitutions in the HA and in some of the other proteins. It was previously hypothesized that an accumulation of multiple basic amino acids at the HACS is the final step required to transform a LPAIV into a HPAIV when the remainder of the viral genome supports a highly pathogenic phenotype [46], which is supported by the results of this study.

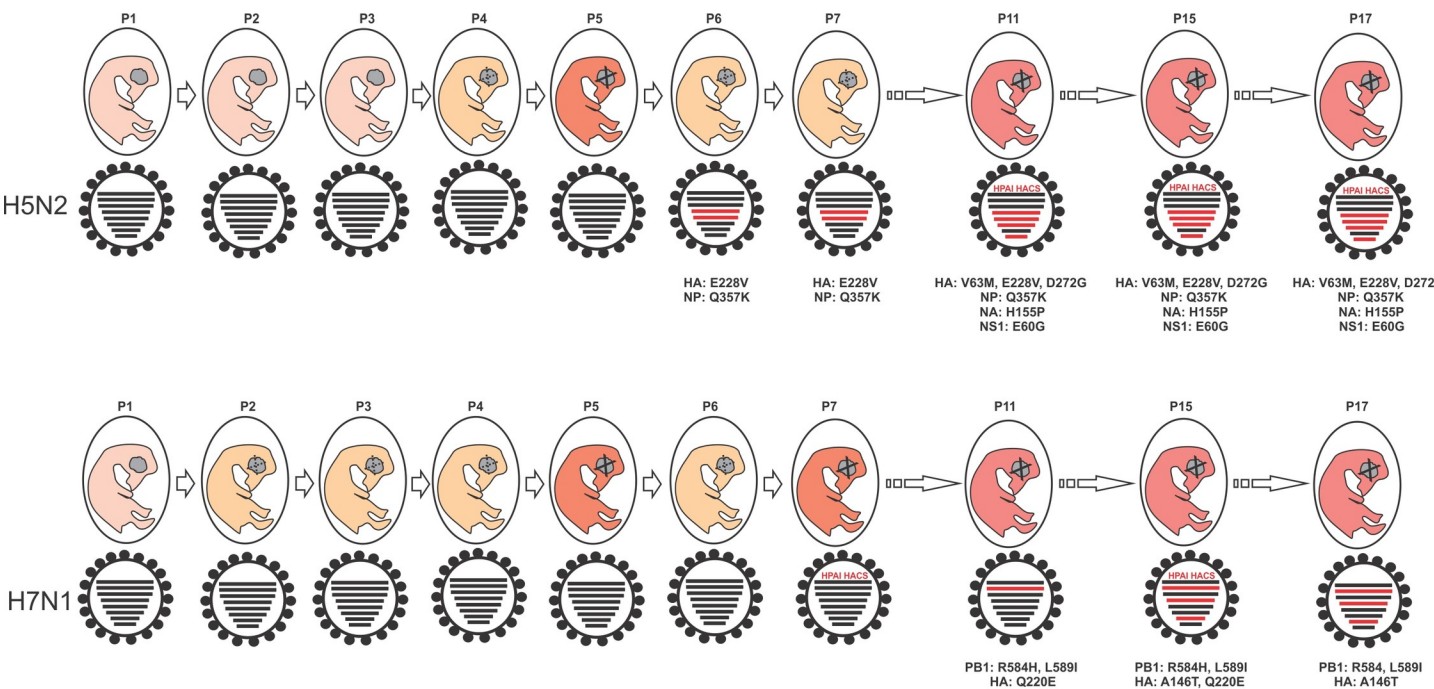

**Fig 2. Schematic overview of the pathogenicity of H5N2 and H7N1 LPAI viruses, the emergence of potential virulence markers and the detection of multi-basic hemagglutinin cleavage sites (MB HACS) in the viral sub-population after seventeen passages in 14-day old ECEs.** Pink embryos indicate mean death times (MDTs) with 100% survival to 90 h, yellow embryos with dotted lines indicate partial survival and/or MDTs between 60 and 90 hours, and red embryos with solid lines indicate 100% mortality within 60 hours or less. Red genome segments indicate one or more amino acid substitutions in the consensus protein sequence with key amino acid substitutions listed below the figures.

Non-homologous recombination is a rare event in RNA viruses [47], but cases of insertions into the HACS have been reported for H7 strains, with the earliest reports stemming from viruses passaged under experimental conditions. Twelve passages of an H7N3 virus [48] and five passages of an H7N7 virus [49] in chicken embryo cells without the addition of trypsin, led to the insertion in the HACS of a 54 nt insert derived from the 28S rRNA gene or nt 284–343 of the NP, respectively, with both viruses demonstrating an increased pathogenicity in chickens. The first reported field case of non-homologous recombination at the HACS was during an H7N1 outbreak in Italian poultry in 1999, with peptide insertions SRVR and SRMR of unknown origin [50]. The SRVR peptide insertion recurred in an H7N3 outbreak in commercial turkeys in 2002 in the Netherlands. The virus had an increased intravenous pathogenicity index in chickens and the authors speculated that the 12-nt insertion was possibly derived from turkey major histocompatibility complex B locus RNA [51]. In 2002, an HPAI H7N3 emerged in in broiler breeder chickens in Chile, that had a 10 aa insert at the HACS corresponding to nt 1268–1297 of the NP gene [11]. An H7N3 virus with increased pathogenicity that caused an epidemic in chicken broiler breeder farms, in 2004 in British Columbia, Canada was found to have a 21 insert of nucleotides 737 to 757 of the M1 gene [52], and another H7N3 HPAI virus in 2007 with increased mortality in 24-week roosters, Saskatchewan, Canada had an HACS insert of TKPRPR of undetermined origin. Yet another H7N3 HPAI virus caused poultry farm outbreaks in Jalisco State, Mexico, and that strain had an 8-amino acid insertion of host 28S rRNA at the HACS [53]. Non-homologous recombination was not detected in the H7N1 virus used in the present study at any passage, however, in the H5N2 virus a 13 nt insert of unknown origin was detected at P6. More significantly, in P17 a 42-nt insert encoding the

peptide RESRNPGNAEIED appeared to be derived from nucleotides 729 to 776 of the NP gene. This is the first report of non-homologous recombination in the HACS of an H5 virus.

Most multi-basic cleavage sites in H5 and H7 viruses in nature contain stretches of between 5 and 8 basic amino acids, and a higher number of basic amino acids correlates with increased pathogenicity [37, 43]. The mechanism of extension of the HACS has only been studied for H5 viruses thus far, entailing the slippage of the viral RNA-dependent RNA polymerase complex, preceded by random point mutations that destabilize the RNA secondary structure adjacent to the HACS [9, 10]. Furthermore, there seems to be a selection bias towards longer multibasic insertions, where mid-length pHACS are rapidly replaced by extended forms [9, 54]. We did not observe any progressive extension of the HACS caused by polymerase slippage as the passages progressed, but seventeen passages may have been insufficient to generate these from a native LPAI progenitor. Most other experimental trials that investigated the conversion of LPAI to HPAI *in vitro*, *in vivo* or *in ovo* used LPAI precursor viruses that were isolated prior to HPAI emergence in the flock, therefore it's difficult to exclude the possibility of the selection of minority HPAI variants already present in the field isolate [14, 37, 50]. This study has provided further insight into how HPAI viruses emerge from low pathogenic precursors but it also demonstrated the pathogenic potential of H5N2 and H7N1 strains that have not yet been implicated in HPAI outbreaks.

## Supporting information

**S1 Table. Ion Torrent sequencing results and read coverage for the H5N2 genome.** Genes encoded within each segment are in italics; nt: nucleotides.
(DOCX)

**S2 Table. Ion Torrent sequencing results and read coverage for the H7N1 genome.** Genes encoded within each segment are in italics; nt: nucleotides.
(DOCX)

**S3 Table. Variants detected at the hemagglutinin cleavage site (HA$_0$) of H5N2 low pathogenic avian influenza viruses passaged in 14-day old embryonated chicken eggs.** [#]Insertions, deletions and substitutions in relation to the conventional low pathogenic sequence[‡] are underlined; stop codons are indicated by "*". [†]In-frame HA$_2$ [] total number of variants: total number of HACS reads ratio.
(DOCX)

**S4 Table. Variants detected at the hemagglutinin cleavage site (HA$_0$) of H7N1 low pathogenic avian influenza viruses passaged in 14-day old embryonated chicken eggs.** [#]Insertions, deletions and substitutions in relation to the conventional low pathogenic sequence[‡] are underlined; stop codons are indicated by "*". [†]In-frame HA$_2$. [] total number of variants: total number of HACS read ratio.
(DOCX)

## Acknowledgments

Virus strains were provided by Deltamune (Pty) Ltd (Pretoria, South Africa). The authors thank Karen Ebersohn for performing virus titrations and Carel Van Heerden and Alvera Vorster for Ion Torrent sequencing services.

## Author Contributions

**Conceptualization:** Celia Abolnik.

**Formal analysis:** Agnes Tinuke Laleye, Celia Abolnik.

**Funding acquisition:** Celia Abolnik.

**Investigation:** Agnes Tinuke Laleye, Celia Abolnik.

**Methodology:** Agnes Tinuke Laleye, Celia Abolnik.

**Project administration:** Celia Abolnik.

**Supervision:** Celia Abolnik.

**Writing – original draft:** Agnes Tinuke Laleye, Celia Abolnik.

**Writing – review & editing:** Celia Abolnik.

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
