## [Decision Letter · Decision Letter 0]

5 Aug 2020

PONE-D-20-15349

Emergence of highly pathogenic H5N2 and H7N1 influenza A viruses from low pathogenic precursors by serial passage in ovo

PLOS ONE

Dear Dr. Abolnik,

Thank you for submitting your manuscript to PLOS ONE. After careful consideration, we feel that it has merit but does not fully meet PLOS ONE’s publication criteria as it currently stands. Therefore, we invite you to submit a revised version of the manuscript that addresses the points raised during the review process.

Please consider carefully comments of reviewer 1, in particular the ones related to the statistical analysis. It seems that probably some technical terms were misused and to the organization of the result and discussion sections.

We look forward to receiving your revised manuscript.

Kind regards,

Maria Serena Beato

Academic Editor

PLOS ONE

Journal Requirements:

2. We noted in your submission details that a portion of your manuscript may have been presented or published elsewhere.

"Some of the results were published in the PhD dissertation of Dr A.T Laleye (University of Pretoria, 2018)."

Please clarify whether this publication was peer-reviewed and formally published. If this work was previously peer-reviewed and published, in the cover letter please provide the reason that this work does not constitute dual publication and should be included in the current manuscript.

6. Please upload a copy of Figures 2 and 3, to which you refer in your text on pages 17 and 21. If the figure is no longer to be included as part of the submission please remove all reference to it within the text.

Reviewers' comments:

Reviewer's Responses to Questions

**Comments to the Author**

1. Is the manuscript technically sound, and do the data support the conclusions?

Reviewer #1: Yes

Reviewer #2: Yes

2. Has the statistical analysis been performed appropriately and rigorously? 

Reviewer #1: No

Reviewer #2: N/A

3. Have the authors made all data underlying the findings in their manuscript fully available?

Reviewer #1: Yes

Reviewer #2: Yes

4. Is the manuscript presented in an intelligible fashion and written in standard English?

Reviewer #1: Yes

Reviewer #2: Yes

5. Review Comments to the Author

Reviewer #1: In the manuscript “Emergence of highly pathogenic H5N2 and H7N1 influenza A viruses from low pathogenic precursors by serial passage in ovo”, the authors used embryonated eggs to serially passage two low pathogenic avian influenza viruses to study the evolution of the phenotype towards a highly pathogenic one, and to correlate the phenotype with substitution appearing in the genome. The manuscript is well written and the study design is rigorous. The work performed is original in the way the substitutions at the cleavage site were studied. Extracting the reads of Next-Generation Sequencing that specifically cover the cleavage site in the heamagglutinin is a nice approach that required probably a huge analytical work.

Introduction

Line 62: the canonical LPAI cleavage motif could be given here to illustrate the positions of the basic amino acids.

Lines 101-107: Are these lines useful in the introduction? Aim of paper is not to explain why 14-day embryos are more susceptive to HPAI selection.

Lines 129-133: This might fit better in either the discussion or the introduction.

Line 142: Rephrase. Mean death times were calculated, but ‘dead’ or ‘alive’ was recorded daily.

Line 150: This change in the protocol is not discussed or explained anywhere in the manuscript. Is it linked to Lines 101-107?

Line 199: Table 1 is giving nucleotide sequences, so it is confusing to see only analyses on amino-acids mentioned. Was any analysis at the nucleotide level performed? Any interesting synonymous substitutions in addition to the non-synonymous, for example in the non-coding regions? This could perhaps be mentioned in the discussion?

Results

Table 2: Maybe transform into figures that might be better for a clearer visualization of the number of positive (out of tested) and MDT?

Generally speaking, there are a lot of discussion elements in the results. Do the authors consider the description of identified mutations by other studies or in samples elsewhere as results or discussion?

Line 282: P11? There is no P10 in table 3. Clarify. In addition, was any sequencing of intermediate passages (between 7 and 11) performed for these mutations? Was I464N found as a difference with the inoculum? Can H150P be considered as a replacement of I464N.

From page 15, there is no line numbering

For H7N1, same comment as for H5N2: elements of discussion are given with the results.

P15 L6: ‘in PB1’ should be mentioned along the 2 substitutions when first mentioned.

“A146T emerged between P8 and P15”: where does this come from? It should be clearer how this data was obtained as it is not presented in Table 4. It seems that, for HA paragraph, the authors took into account the “quasispecies” to evaluate when the substitution appeared. It should be clearer in the text and table 4.

For M1, not clear. Only one substitution mentioned in table 4.

Table S3 and similar: indication of frameshift with amino-acid consequences in the cleavage site. But what about full length HA: truncated forms in addition to changes in cleavage site? What does it mean “in frame HA2”? Does it mean that for the other variants, this is not the case? Clarify.

In the text, highlight (underline or bold) also the differences in the cleavage sites to facilitate the reader’s understanding.

Data of this cleavage site analysis should be synthesized in a clearer manner as it is not easy to follow the proportions of the variants over passages. These proportions should be given in the text. This is what makes a variant potentially relevant. See also comments over the Discussion, as this needs to be discussed in light of the Ion Torrent error rate.

Discussion:

Lines 9-10: deep sequencing on original sample or of the stock used for the experiments reported in this manuscript (passage 3 in 9-10 d old embryonated eggs)?

Line 13: significantly? There are no statistic test results.

Lines 21-22: “markers in the proteins encoded by the consensus sequences” is not clear.

For the identified mutations: have they been tested alone or in combination using reverse genetics to study their impact? This could be mentioned and discussed.

The results of the cleavage site analysis should be discussed in light with the level of error rate of Ion Torrent. Which proportion of a variant was taken as “true”? When there are only very few reads concerned with a substitution/deletion, is it relevant? This should be discussed.

The authors should try to find a way to summarize the finding of a figure. They talk about correlation between substitutions and pathogenicity, but Figure 1 comes too late and does not present the specific mutations that the authors suggest as marker of pathogenicity.

Conclusions might need to be slightly amended based of analysis of proportions and taking into account error rate of technique.

Reviewer #2: Thank you for allowing me to review the paper by Abolnik and colleague entitled "Emergence of highly pathogenic H5N2 and H7N1 influenza A viruses from low pathogenic precursors by serial passage in ovo". The authors present data on H5N2 and H7N1 LPAI naturally occurring influenza viruses in ostrich; and serially passaged these isolates in eggs to force emergence of mutations that correlated with higher pathogenicity. A few comments:

Around passage 8-9 for both viruses (Table 2) seems to be a switch of phenotype for both viruses, as all embryos were dying up to passage 7 but then switches to 100% live and an increase on mean death time (MDT), is there an explanation for this? May be include a brief comment or explanation. Could it be protocol related?

Text needs some formatting: for example, line numbering stops in page 13 and start again in the discussion section, also it is confusing to find figure 1 legend in the middle of the discussion section (page 28).

Overall the paper gives clear results and interesting observations without overstating their findings.

6. PLOS authors have the option to publish the peer review history of their article (what does this mean?). If published, this will include your full peer review and any attached files.

Reviewer #1: No

Reviewer #2: No

---

## [Author Response · Author response to Decision Letter 0]

10 Sep 2020

PONE-D-20-15349

Response to Reviewers’ comments

Reviewer #1 

In the manuscript “Emergence of highly pathogenic H5N2 and H7N1 influenza A viruses from low pathogenic precursors by serial passage in ovo”, the authors used embryonated eggs to serially passage two low pathogenic avian influenza viruses to study the evolution of the phenotype towards a highly pathogenic one, and to correlate the phenotype with substitution appearing in the genome. The manuscript is well written and the study design is rigorous. The work performed is original in the way the substitutions at the cleavage site were studied. Extracting the reads of Next-Generation Sequencing that specifically cover the cleavage site in the heamagglutinin is a nice approach that required probably a huge analytical work.

INTRODUCTION

Line 62: the canonical LPAI cleavage motif could be given here to illustrate the positions of the basic amino acids.

Line 62 was modified to “Typical LPAI viruses contain the HA0 motifs PQRETR/GLF for H5 or PELPKGK/GLF for H7, i.e. single or non-consecutive basic amino acids (R or K) adjacent to the cleavage site”

Lines 101-107: Are these lines useful in the introduction? Aim of paper is not to explain why 14-day embryos are more susceptive to HPAI selection.

The paragraph was included to facilitate the reader’s understanding of the choice of 14-day old embryonated eggs for the study. No change made. 

Lines 129-133: This might fit better in either the discussion or the introduction. 

The text was moved to the Introduction as suggested

Line 142: Rephrase. Mean death times were calculated, but ‘dead’ or ‘alive’ was recorded daily.

Line 142 was rephrased as suggested

Line 150: This change in the protocol is not discussed or explained anywhere in the manuscript. Is it linked to Lines 101-107?

Yes. To clarify this the sentence was modified as follows: “Passages 1 to 7 were performed using only the aspirated AFs but, to ensure the broadest possible representation of the viral population within the embryo, from passage eight onwards the whole embryos were harvested along with the AF…”

Line 199: Table 1 is giving nucleotide sequences, so it is confusing to see only analyses on amino-acids mentioned. Was any analysis at the nucleotide level performed? Any interesting synonymous substitutions in addition to the non-synonymous, for example in the non-coding regions? This could perhaps be mentioned in the discussion?

Table 1 details the various modified sequence tags (MSTs) we used to retrieve the HA0-spanning regions (subsequently translated to amino acids), and the canonical amino acid sequence is provided for reference. No, we did not analyze the synonymous substitutions due to the sheer volume of the data we generated. No changes made. 

RESULTS

Table 2: Maybe transform into figures that might be better for a clearer visualization of the number of positive (out of tested) and MDT?

Table 2 was converted to a figure (Figure 1) as suggested. Table and figure numbers were adjusted accordingly throughout the manuscript. 

Generally speaking, there are a lot of discussion elements in the results. Do the authors consider the description of identified mutations by other studies or in samples elsewhere as results or discussion?

Yes we did consider it and after various drafts of the manuscript we decided that the present format was the most clear and concise for readers. No changes made.

Line 282: P11? There is no P10 in table 3. Clarify. In addition, was any sequencing of intermediate passages (between 7 and 11) performed for these mutations? Was I464N found as a difference with the inoculum? Can H150P be considered as a replacement of I464N.

The typing error has been addressed; P10 was corrected to P11 in Table 3 (renamed as Table 2). H150P is also a typing error; it was corrected to H155P in the table and text. It might be possible that H155P is a compensation for I4464N, but functional studies would be required to verify this. 

From page 15, there is no line numbering

Corrected in the revised version.

For H7N1, same comment as for H5N2: elements of discussion are given with the results

As above, the discussion points in the Results section are pertinent to the specific mutations we observed, and to remove them here and incorporate into the Discussion would mean the reader is constantly paging back and forth in reference to the tables. The Discussion would then be extremely long and verbose, whereas the Results section would contain only the tables. We would like to retain the manuscript in its present format as this makes it the easiest for the reader to follow.

P15 L6: ‘in PB1’ should be mentioned along the 2 substitutions when first mentioned.

Changed as suggested.

“A146T emerged between P8 and P15”: where does this come from? It should be clearer how this data was obtained as it is not presented in Table 4. It seems that, for HA paragraph, the authors took into account the “quasispecies” to evaluate when the substitution appeared. It should be clearer in the text and table 4.

A146T was/is presented in Table 4 (now Table 3), no change made. 

For M1, not clear. Only one substitution mentioned in table 4.

The only substitution (D231N) observed at consensus level in the M1 is the one presented in the Table 4 (now Table 3), no change made.

Table S3 and similar: indication of frameshift with amino-acid consequences in the cleavage site. But what about full length HA: truncated forms in addition to changes in cleavage site? What does it mean “in frame HA2”? Does it mean that for the other variants, this is not the case? Clarify.

The sequences presented here covers only the cleavage site of the HA and not the entire HA protein. Typically, the HACS starts with proline-encoding codon and terminates in Phenylalanine-encoding codon. As we indicated in Table S3, insertions or deletions can cause +1 or +2 frameshifts in the open reading frame (ORF) of the HA2 gene, leading to truncation of the protein. Sequences that produce the correct ORF are said to be “in-frame”. The latter is a widely-used term. No changes made.

In the text, highlight (underline or bold) also the differences in the cleavage sites to facilitate the reader’s understanding.

The differences in the HACS sequences for both H5 and H7 are now underlined in the text with “(variation underlined)” added

Data of this cleavage site analysis should be synthesized in a clearer manner as it is not easy to follow the proportions of the variants over passages. These proportions should be given in the text. This is what makes a variant potentially relevant. See also comments over the Discussion, as this needs to be discussed in light of the Ion Torrent error rate.

This manuscript represents the analysis of a massive amount of data and a lot of thought was given to what the best way would be to make it concise and interesting to a reader, yet avoid the pitfalls of over-interpretation. The proportions of the variants between passages (including those in the HACs) will be directly related to the depth of coverage we obtained for each segment (Tables S1 and S2) as well as the depth of coverage of the reads in specific regions of the segment that tends to be highly variable (data not shown). In some cases the coverage was very low, and this would certainly cause problems in assigning relevance to the proportions of variants. Therefore, to avoid over-interpreting the data, we focused on the cumulative effects of the mutations, and only when a variant emerged in the consensus genome was it flagged as being potentially relevant. We have made all the raw sequence data publically available should follow-up studies be of interest to anyone. 

DISCUSSION

Lines 9-10: deep sequencing on original sample or of the stock used for the experiments reported in this manuscript (passage 3 in 9-10 d old embryonated eggs)?

Both, although on the originals stocks the aligned HACs was only examined visually in the CLC Genomics Workbench (not results of this study). For clarification the sentence was modified to “…and deep sequencing of the stocks used for the experiments confirmed that only LPAIVs were present in the sub-populations.”

Line 13: significantly? There are no statistic test results.

“significantly” was replaced with “markedly”

Lines 21-22: “markers in the proteins encoded by the consensus sequences” is not clear.

This refers to mutations in viral proteins other than the HA. To clarify this, the text was rephrased to read “molecular markers in the various proteins encoded by the consensus sequences”

For the identified mutations: have they been tested alone or in combination using reverse genetics to study their impact? This could be mentioned and discussed.

No study has been carried out to determine the impact of the novel mutations observed. However, this could be carried out in the future as was recommended in parts of the discussion

The results of the cleavage site analysis should be discussed in light with the level of error rate of Ion Torrent. Which proportion of a variant was taken as “true”? When there are only very few reads concerned with a substitution/deletion, is it relevant? This should be discussed.

A high Phred cutoff score (20) was applied to filter reads prior to analysis (as per the Materials and Methods section). We also stated in the manuscript that the variants we detected were probably an underrepresentation of what was present in the population. The following was added to the discussion: “Ion Torrent sequencing is known to have a high error rate in base calls in long homopolymer regions (45), but extended homopolymeric regions such as those caused by RNA polymerase slippage were absent from our data. The reference list was updated with Besser et al., 2018. 

The authors should try to find a way to summarize the finding of a figure. They talk about correlation between substitutions and pathogenicity, but Figure 1 comes too late and does not present the specific mutations that the authors suggest as marker of pathogenicity.

The figure (renamed as Fig. 2) has been modified to list the key amino acid substitutions from Tables 2 and 3 on it. 

Conclusions might need to be slightly amended based of analysis of proportions and taking into account error rate of technique.

Addressed in previous comments

Reviewer #2

Thank you for allowing me to review the paper by Abolnik and colleague entitled "Emergence of highly pathogenic H5N2 and H7N1 influenza A viruses from low pathogenic precursors by serial passage in ovo". The authors present data on H5N2 and H7N1 LPAI naturally occurring influenza viruses in ostrich; and serially passaged these isolates in eggs to force emergence of mutations that correlated with higher pathogenicity. A few comments:

Around passage 8-9 for both viruses (Table 2) seems to be a switch of phenotype for both viruses, as all embryos were dying up to passage 7 but then switches to 100% live and an increase on mean death time (MDT), is there an explanation for this? May be include a brief comment or explanation. Could it be protocol related?

Yes the reviewer is correct, we did notice this and the following text was added to provide an explanation: “The percentage of live embryos and the MDTs had been decreasing up until passage 7, but in passage 8 the phenotype of both viruses changed with an increase in the percentage of live embryos and MDTs of >90 hours. The only change in the protocol was that stocks were frozen at -80�C during a University recess; this likely caused a slight drop in the viability of the viruses in passage 8, causing the delayed embryo deaths.”

Text needs some formatting: for example, line numbering stops in page 13 and start again in the discussion section, also it is confusing to find figure 1 legend in the middle of the discussion section (page 28).

Line numbers were included through the length of the original manuscript uploaded, some formatting might have been lost due to different versions of Microsoft office. This is now corrected.

Overall the paper gives clear results and interesting observations without overstating their findings.

Thank you.

---

## [Decision Letter · Decision Letter 1]

24 Sep 2020

Emergence of highly pathogenic H5N2 and H7N1 influenza A viruses from low pathogenic precursors by serial passage in ovo

PONE-D-20-15349R1

Dear Dr. Abolnik,

We’re pleased to inform you that your manuscript has been judged scientifically suitable for publication and will be formally accepted for publication once it meets all outstanding technical requirements.

Kind regards,

Camille Lebarbenchon

Academic Editor

PLOS ONE

Additional Editor Comments (optional):

Reviewers' comments:

Reviewer's Responses to Questions

**Comments to the Author**

1. If the authors have adequately addressed your comments raised in a previous round of review and you feel that this manuscript is now acceptable for publication, you may indicate that here to bypass the “Comments to the Author” section, enter your conflict of interest statement in the “Confidential to Editor” section, and submit your "Accept" recommendation.

Reviewer #1: (No Response)

2. Is the manuscript technically sound, and do the data support the conclusions?

Reviewer #1: Yes

3. Has the statistical analysis been performed appropriately and rigorously? 

Reviewer #1: N/A

4. Have the authors made all data underlying the findings in their manuscript fully available?

Reviewer #1: Yes

5. Is the manuscript presented in an intelligible fashion and written in standard English?

Reviewer #1: Yes

6. Review Comments to the Author

Reviewer #1: The new Figure 1 is still in fact a table. If a true figure is not made, then go back to a true table that is formatted according to the journal requirements

Table 3: I am still confused by A146T. It is indeed presented in table 3, but only at P15 and P17. But the text still states that it was detected from P8. So why is A146T also indicated for P11 in table 3?

I still think that there is too much discussion in the results.

7. PLOS authors have the option to publish the peer review history of their article (what does this mean?). If published, this will include your full peer review and any attached files.

Reviewer #1: No

---

## [Editor Report · Acceptance letter]

28 Sep 2020

PONE-D-20-15349R1 

Emergence of highly pathogenic H5N2 and H7N1 influenza A viruses from low pathogenic precursors by serial passage in ovo 

Dear Dr. Abolnik:

I'm pleased to inform you that your manuscript has been deemed suitable for publication in PLOS ONE. Congratulations! Your manuscript is now with our production department. 

Kind regards, 

on behalf of

Dr. Camille Lebarbenchon 

Academic Editor

PLOS ONE